# Coffee (*Coffea arabica* L.): Methods, Objectives, and Future Strategies of Breeding in Ethiopia—Review

**Yebirzaf Yeshiwas Melese [1,\*] and Semagn Asredie Kolech [2]**

[1] Department of Horticulture, College of Agriculture and Natural Resources, Debre Markos University, Debre Markos P.O. Box 269, Ethiopia
[2] Amhara Regional Agricultural Research Institute, Bahir Dar P.O. Box 527, Ethiopia; sekol2009@gmail.com
\* Correspondence: Yebirzaf_Yeshiwas@dmu.edu.et

**Abstract:** *Coffea arabica* L. belongs to the *Rubiaceae* family, and the genus *Coffea* is believed to have a primary center of origin and genetic variability in the highlands of southwestern Ethiopia. It is a vital beverage commodity across the world and a valuable export product, ranking second in international trade after petroleum. Ethiopia is among the top five major coffee-producing countries and is Africa's leading producer. However, its full production capacity has not yet been exploited, and research efforts to reduce biotic and abiotic factors through reproduction have been extremely limited. Hence, improvement through different breeding methods is essential to overcome the constraints in its production. Thus, the objective of this study is to review the different breeding methods applied for different traits in Ethiopia. Breeding methods depend on the type and the source of traits and the final breeding objectives. The main breeding objectives are production, resistance/tolerance to diseases, and cup quality. The commonly applied breeding methods are selected and intra-specific hybridization, germplasm enhancement, and the development of improved varieties with wider adaptability. There is also a practice of crossing parental lines selected for certain desirable traits for the development of hybrid varieties. Accordingly, some promising success has been obtained. Forty-one coffee varieties have been released so far. Because conventional breeding methods are time-consuming, integrating conventional breeding methods with biotechnological techniques could have an instrumental role in the rapid development of suitable varieties for the changing climate.

**Keywords:** coffee; genetic diversity; coffee breeding; improved varieties; hybrid varieties

## 1. Introduction

Coffee (*Coffea arabica* L.) belongs to the family *Rubiaceae*, and the genus *Coffea* is mostly grown in subtropical and tropical regions [1] and consists of 90 to 124 species [2,3]. *Coffea canephora* Pierre and *Coffea arabica* L. are the only two species that are economically important and widely cultivated across the world. *Coffea liberica* is also cultivated on a small scale to satisfy local consumption [4]. The primary center of origin and genetic diversity for *Coffea arabica* L. is the highlands of southwestern Ethiopia, where it occurs naturally in the undergrowth of the Afromontane rainforests between 1000 and 2000 m above sea level [1,5–10].

Almost all coffee species are diploid (2n = 2x = 22) and most are self-incompatible, except for *Coffea arabica*, which is a self-fertile species and a natural allotetraploid (2n = 4x = 44). *Coffea arabica* is a self-pollinating species with a common outcrossing rate of less than 10%, which is sufficient to induce some variation in offspring and free-pollinating cultivars [1].

Coffee is one of the world's most important beverage crops and a valuable agricultural export commodity; it is the second-most commonly traded commodity in the world, after oil [11]. Arabica coffee represents 70% of the world's coffee production [12]. It is cultivated in over 80 countries and covers over 10.2 million hectares of land in the tropical and subtropical regions of the world, particularly in Africa, Asia, and Latin America [13].

Globally, the coffee trade generates about USD 10–12 billion annually for the producing countries and provides job opportunities for some 20–25 million people, who grow, process, distribute, and market the product [14–17].

In 2016, from the total national annual export revenues, coffee represented a 10% increase in earnings for Brazil (USD 4.84 billion) [18].

Ethiopia is one of the five largest coffee-producing countries in Africa [19]. The percent share of Ethiopian coffee to the world coffee market in four consecutive years has been 4.1, 4.1, 4.4, and 4.3 in 2013, 2014, 2015, and 2016, respectively [20]. In Ethiopia, coffee accounts for 60–70% of the national foreign exchange income and the livelihoods of some 20–25% of the population, directly or indirectly [21]. During the 2018/19 Meher Season, 764,863.16 ha of land was allotted for coffee production and 494,574.36 tones were obtained, with average productivity of 0.64 tones/ha$^{-1}$ [22,23]. According to Tefera and Bickford [24] coffee exports reached 248,129 MT valued at USD 821,140.00 during 2019/20.

Despite its importance, the presence of great genetic diversity, and diverse agronomic species and suitable soils, coffee production in Ethiopia is lower compared to high-yielding countries such as Brazil, Colombia, Indonesia, and Vietnam. The estimated yield is less than 200 kg ha$^{-1}$ for forest coffee and around 450–750 kg ha$^{-1}$ for semi-modern coffee plantations [23], which is less by far than the 2443 kg ha$^{-1}$ yield obtained by Brazil [25]. Arabica coffee production is affected by poor crop management practices, low yields, poor soils, diseases, pests, limited access to market information, lack of physical infrastructure, lack of improved hybrids, poor extension services, and climate change [26–28].

The Arabica coffee grown in Ethiopia is organic, fertilizers, pesticides, and herbicides are not applied, and the coffee is grown wild in the forests. Additionally, the seedlings used during planting are not grafted and need frequent and timely pruning. Similarly, the development of a disease-resistant and high-yielding variety by the traditional breeding approach is lengthy and inefficient. The multiplication of high-yielding, disease-free seedlings depends solely on hand pollination and, to a lesser extent, on advanced breeding methods such as clonal propagation by cutting and grafting. This is due to delays in protocol optimization for mass multiplication using tissue culture techniques [29]. Other major coffee producing countries have widely applied molecular breeding approaches to introduce new traits into selected coffee genotypes and develop new cultivars with desirable traits and a high yield [13].

Moreover, Ethiopia's full production capacity has not been exploited, and research and breeding efforts are limited and insufficient to respond to the diverse agroecologies of the country [30]. Consequently, the yield per hectare remains too low, at 0.673 tons ha$^{-1}$ [31]. The development and improvement of high-yield and stress-tolerant coffee varieties are the main objectives for conventional and molecular breeding methods. Hence, it is evident that the improvement of coffee through different breeding methods is essential and is a critical task to address in order to overcome the constraints of coffee production. There have been research studies conducted over the last two to three decades in Ethiopia to improve coffee in terms of important traits such as productivity, resistance to disease and insect pests, and quality. Thus, reviewing and summarizing the recent research output on coffee breeding will help scientists to identify the research gaps and propose a future line of work for exploiting the potential of the crop. Therefore, the objectives of this review are:

- Assess the different breeding methods applied, identify the research gaps, and highlight future strategies for coffee breeding in Ethiopia.

## 2. Coffee Growth, Floral Biology, and Fruit Characters

### 2.1. Coffee Growth

*Coffea arabica* is a dicotyledon evergreen that reaches 8 to 10 m in height. Each node produces two, opposite leaves and therefore has two leaf axils residing on opposite sides of the node, each containing a series of buds. Branching is dimorphic as there are two kinds of buds: 'serial buds' and 'head of series buds'. The head of series buds develop into 'plagiotropic' (horizontal) branches, whilst the lower serial buds produce more 'orthotropic'

(vertical) shoots, commonly called 'suckers', when the tip of the main orthotropic shoot is damaged. The shoot system of the coffee plant is illustrated in Figure 1. Inflorescences develop from serial buds at each node on the plagiotropic branches [32–34].

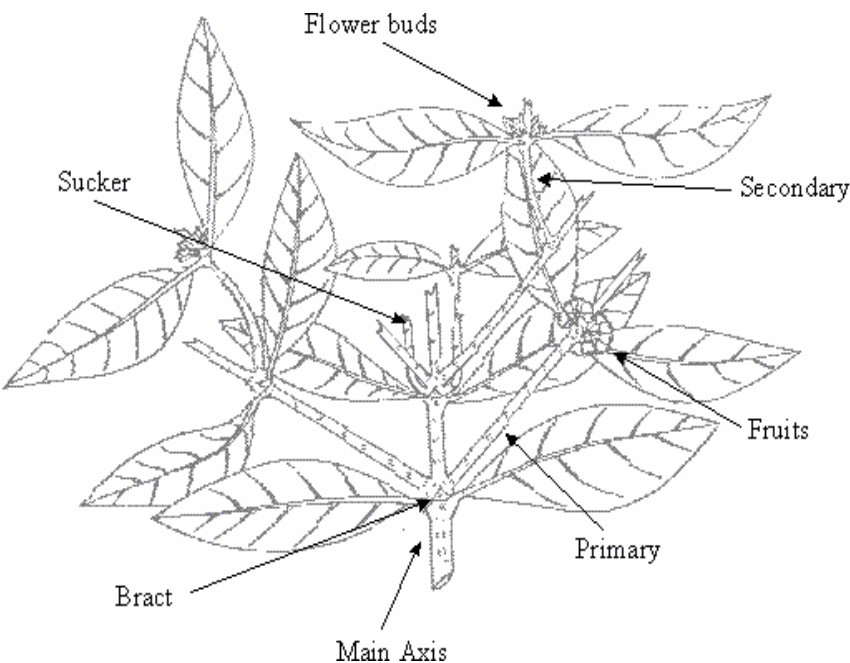

**Figure 1.** Morphology of mature Arabica coffee plant; Source: Obso [35].

*2.2. Floral Biology and Fruit Characters*

After 3–4 years of transplanting, fragrant white flowers grow in clusters in the axils of coffee leaves. The inflorescence has a short axis, two pairs of bracts at its base, and varying in number from one to twenty per leaf axil on the primary and secondary branches. The corolla is white and has five expanded petals. The five stamens are epipetalous and inserted in the corolla tube between the petals on short filaments; this facilitated emasculation. The anthers are bilocular, opening vertically/longitudinally. The pollen grains are numerous and small in size. Under normal conditions, pollen loses its viability very rapidly [36]. It is, however, possible to keep the viability for more than two years by storing it under vacuum conditions at −18 °C. The ovary is inferior with a long terminal style and two stigmatic branches and is made up of two united carpels with one ovule per carpel [37].

The flowering of the coffee plant involves two separate processes, bud initiation and flower opening, also known as antithesis. The coffee buds that will develop into flowers are usually produced 4 to 5 months before anthesis. Initiation of the flower buds occurs when the serial buds on plagiotropic branches are induced to differentiate into floral buds. The floral buds open on sunny days in the early morning, and pollen removal begins soon after. The style is receptive when the bud opens and remains receptive for three to four days depending on climatic conditions [38]. However, Walyaro and Van der Vossen [37] report that stigmas are receptive for at least nine days and recommend that the vesicles should not be removed for two weeks after pollination for successful hybridization.

Buds can reach 4 to 6 mm and then enter a dormancy period. Dry periods are necessary to break the dormancy of floral buds and their final development and fruiting are often triggered by rainfall after a dry period. This occurs because after several weeks of water stress, rain stimulates floral growth, which will then open in eight to ten days. Fertilization takes place before or just at flower opening and then after pollination, where the fusion of one male nucleus and the polar nuclei forms an endosperm, and this endosperm then forms the coffee bean [39]. About 6 to 8 weeks after each coffee flower is fertilized, cell division occurs and the coffee fruit remains as a pinhead for a climate-dependent period of time.

Likewise, Cannell [40] explained that phytohormone levels in the coffee plant were affected by an extended dry season. During the first 3 to 4 days after a water stimulus, meiosis cell division occurs and there is an increase in the levels of endogenous, active, gibberellic acid in the floral bud. Coffee flowers are ephemeral, usually only lasting for two days.

According to Carvalho and Monaco [41], Van der Vossen [42], Arabica coffee is the only autogamous (self-compatible) species of the genus *Coffea* with only up to 10% of natural cross-pollination, whereas all other species are allogamous with gametophytic self-incompatibility. Pollination is affected by both wind (anemophily) and insects (entomophily). Having 10% natural cross-pollination could be adequate to retain heterozygosity within the population to a certain level. Therefore, selfing for one or two generations is important to guarantee sufficient homozygosity in parental lines destined to be used in a crossing program. William [43] reported that at Jimma Agricultural Research Center 10–40% outcrossing was recorded. Bud flower emasculations were performed 2–3 days before anthesis. Pollination was performed during the next day of emasculation. Labeling was conducted with the recording of details of the male and female parents, the date of emasculation, the date of pollination, and the name(s) of the breeders.

Prado et al. [44] explain that several floral morphological traits differ significantly between species and farm types. *C. arabica* plants grown in the shade had a corolla diameter 1.4% larger and an anthers height of 12.8% greater than those grown in the sun. Only the length of the tube was significantly longer in the sunny plantations; 8.7% longer in the sun than in the shade (Figure 2). In contrast, for *C. canephora*, there was no significant main effect on the type of exploitation of floral reproductive traits.

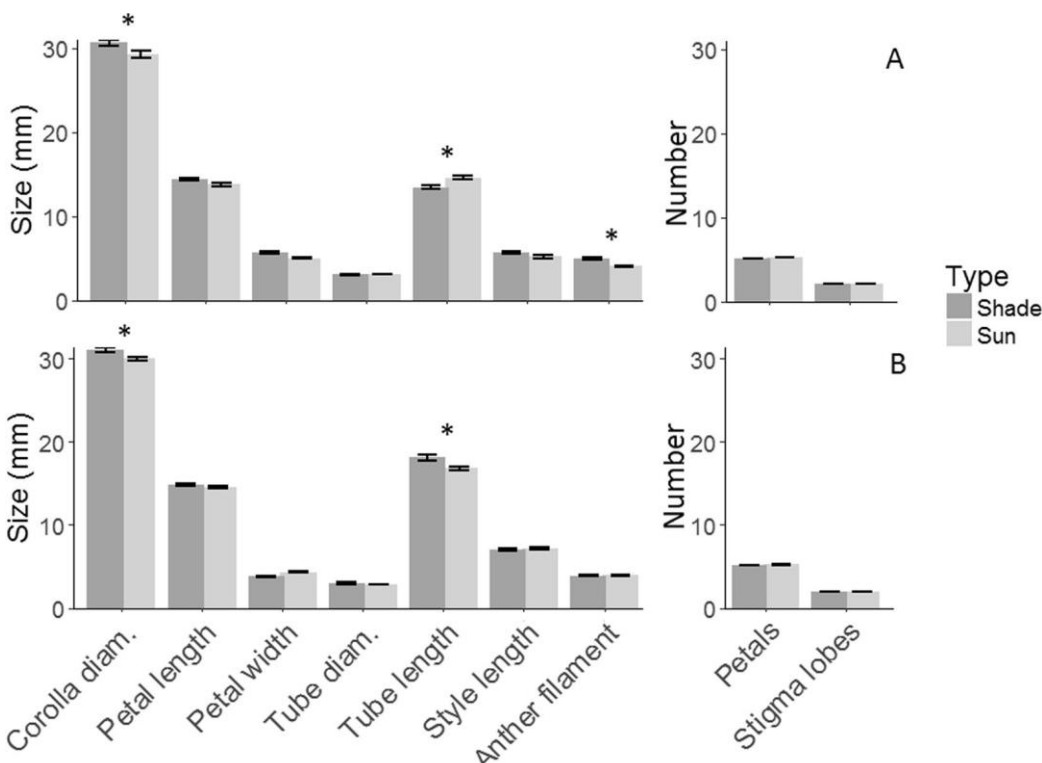

**Figure 2.** Average (±standard error) floral traits of (**A**) *Coffea arabica* and (**B**) *Coffea canephora*. * = significant differences between means of shade and sun at ($p < 0.05$). Source: Prado et al. [44].

Coffee fruit's pericarp is composed of exocarp, mesocarp, and endocarp, in which the seeds are enclosed. Immature berries have a dull green color; however, on ripening the skin color changes from yellow to bright pink. Each coffee berry contains two seeds with a length 8.5–12.5 mm, which are ellipsoidal and pushed together by a flattened surface that is deeply grooved; the outer surface is convex. Thin, silvery testa follow the outline of

endosperm, so fragments are often found in the ventral groove after preparation. Seeds contain mainly green corneous endosperm and a small embryo near the base. Dried beans, after removal of the silver skin, provide the coffee beans for commercial purposes [45].

## 3. Production and Productivity of Coffee in Ethiopia

In Ethiopia, coffee grows between 550 and 2750 m above sea level. However, most Arabica coffee grows best at altitudes between 1300 and 1800 m above sea level. It requires annual rainfall ranges from 1500 to 2500 mm with minimum and maximum ideal air temperatures of 15 °C and 30 °C, respectively. In places where there is less rainfall, such as in the eastern part of the country (about 1000 mm), the coffee is supplemented with irrigation water. It also grows best in the shady environments of Ethiopia's highland forests [21,46,47].

The average yield of green coffee beans per year is 0.7 tons, which is lower than the global average and the Brazilian average 0.8 tons ha$^{-1}$ and 1.3 tons ha$^{-1}$, respectively [48]. Coffee is cultivated by over 4 million small farm holders (Table 1). CSA [49] reported that more farmers cultivate and produce coffee than fruit. Almost 95% of coffee is grown on small plots, often less than half a hectare.

**Table 1.** Estimated number of coffee growers, area coverage, and yield (t/ha) over six years.

| Year | Coffee Growers | Area in ha | Percentage Change in Area | Production in Quintal | Percentage Change in Production | Yield q/ha | % Change in Yield |
|---|---|---|---|---|---|---|---|
| 2012/13 | 4,217,961.00 | 528,751.11 | - | 373,940.642 | - | 0.707 | - |
| 2013/14 | 4,546,785.00 | 538,466.80 | 2.00 | 392,006.222 | 5.00 | 0.728 | 3.00 |
| 2014/15 | 4,723,483.00 | 561,761.82 | 4.00 | 419,980.156 | 7.00 | 0.748 | 3.00 |
| 2015/16 | 5,270,777.00 | 653,909.76 | 16.00 | 414,596.455 | −1.00 | 0.634 | −15.00 |
| 2016/17 | 6,455,194.00 | 700,474.69 | 7.00 | 469,091.124 | 13.00 | 0.67 | 6.00 |
| 2017/18 | 5,019,513.00 | 725,961.24 | 4.00 | 449,229.808 | −4.00 | 0.619 | 8.00 |

Source: Jima Degaga [50].

Ethiopia's coffee production is generally characterized by four production systems. That is forest coffee, semi-forest coffee, garden coffee, and plantation coffee. The main coffee growing areas and denominations as international trade in Ethiopia are Limu, Jimma (commercially Djimma), G(h)imbi and Lekempti, Sidamo, Yirgachefe, Illubabor, Harar, Tepi, and Bebeka. The additional coffee specialty has been recently identified in Amaro, Amhara, Arsi, Balé forest, Borena, Guji, Kaffa forest, Omo, and many other areas [51,52].

## 4. Methods and Objectives of Coffee Breeding in Ethiopia

Different Arabica coffee breeding methods have been applied in different coffee-producing countries around the world with the ultimate goal of improving yield and quality. However, the application of methods may vary depending on the amount of genetic variation available, the ecological conditions, the ultimate breeding goal, and the dominant production issues [42–53].

### 4.1. Breeding Methods

In Ethiopia, the common breeding methods applied to *C. arabica* include adaptability, hybridization, and intraspecific hybridization methods, with an emphasis on improving germplasm and developing improved varieties through breeding for disease resistance and pest tolerance, high yield, quality, and adaptation to various agroecological regions [54]. The different breeding methods are shown in Table 2.

**Table 2.** General breeding methods applied to coffee.

| No. | Method | Source Population | Breeding System | Output | Propagation by | Example |
|---|---|---|---|---|---|---|
| 1 | Arabica Pure line selection | Variety | Selfing | Line | Seed | Caturra (Brazil), Kent (India) SL 28 (Kenya) java (Cameroon) |
| 2 | Intraspecific F1 hybrid | Varieties/accession, the pedigree of crosses | Crossing and selfing | Composite hybrid, F1 hybrid F1 clones | Seed (hand-pollination) Somatic embryogenesis | Ruiro II (Kenya) Ababuna (Ethiopia) in progress Catimor × Et (C. America) |
| 3 | Pedigree selection after hybridization (sometimes also backcrossing) | Varieties | Crossing and selfing | Line | Seed | Catuai, Tupi (Brazil),catimor Sarchimor (costa Rica), S795(India) Colombia (Colombia) |
| 4 | Interspecific hybridization (Arebica × Robusta) backcrossing and pedigree selection | Arebica varieties tetraploid/diploid robusta genotype | Crossing and selfing | Line | Seed | Lacatu Brazil, S2828 (India) |

Source: Clarcke and Vitzthum [55].

### 4.1.1. Selection and Stability Analysis
Genetic Variability

The improvement of *C. arabica* can be divided into several stages, the first of which is the selection and testing of plants that are superior to the existing genetic diversity. Its nature and extent are essential for the development of an effective plant breeding program [56].

Ethiopia is the sole center of origin and diversity of *Coffea arabica*, having great genetic diversity, mainly due to its different agronomic characteristics, such as suitable altitude, ample rainfall, optimum temperature, fertile soils, and the presence of indigenous methods of coffee production in the country [9,57]. Regarding the genetic diversity of *Coffea arabica*, various phenotypic and molecular studies have been conducted in Ethiopia. The results reveal that higher genetic variability is found in Ethiopia than anywhere else in the world [32,58–64]. Thus, the presence of higher genetic variability indicates the potential of *Coffea arabica* for breeding purposes [46,65,66].

The stability of performance in the environment is one of the most desirable properties of genotypes to be recommended for an extended crop. Lemi et al. [67] was study genotypic stability and environmental effects on coffee in eight environments (locations and years) and analyzed Additive main effect and Multiplicative Interaction (AMMI), the AMMI stability value (ASV), the cultivar superiority index (Pi) and the yield stability index. The result indicated that combined analysis of variance exhibited a highly significant difference among the genotypes and genotype by environment interaction. Lashermes et al. [68] and Steiger et al. [69] analyzed the genetic diversity of coffee accessions collected by the FAO and ORSTOM from Ethiopia, which they described as 'subspontaneous-derived accessions', since their identity is not known for certain by using molecular markers. The result revealed the presence of high genetic variability among the Ethiopian coffee populations. Similarly, a study by Mesfin and Bayetta [60] on second progeny Arabica coffee collections of Ethiopian origin shows the prevalence of a high degree of variability in morphological, agronomic, and biochemical traits.

Selection Procedures

The usual breeding steps, such as screening, adaptation trial at different locations, verification of the promising selections, variety release, and seed multiplication, were followed in the pure line coffee variety development breeding program in Ethiopia until 1994. All varieties collected under the national program and the international coffee collection program were rigorously evaluated for a different number of years at each breeding stage [70]. For instance:

Step 1: Mother tree selected with desirable traits from the forest coffee. However, selections are made from the genetically variable original population;

Step 2: Evaluation of mother trees, seedling transplanting, and field management;
Step 3: Replicated yield trials with the remaining lines, especially comparing them with established commercial varieties;
Step 4: Multilocation trials for many years (seasons), usually at 3–5 locations for 3–5 years.
Step 5: Variety release, seed multiplication, and distribution.

However, in cases of urgency, a short-term breeding program, called a 'crash' program, will be employed. The crash breeding program is the employment of the simultaneous execution of multiple stepwise activities. Crash breeding approaches were further improved by employing a special program known as 'local landrace development' which is very useful for maintaining the typical qualities of each area or locality, minimizing adaptation problems, and maintaining farmers' and consumer's preferences. The prior implementation of crash programs has resulted in the production of 13 coffee berry disease (CBD) resistant pure-line varieties in a very short period time when the Ethiopian coffee industry was in severe crisis due to the outbreak of the disease. Considerable progress has also been made in developing 10 additional high-yielding, disease-resistant pure-lines for different agroecologies using the conventional long-term program [71].

### 4.1.2. Hybridization Methods
#### Development of Hybrid Coffee Varieties

In Ethiopia, hybridization is a method in which desirable traits of two or more species and varieties are combined or transfer from one species to another. In the Ethiopian *C. arabica* hybridization development program, the initial breeding objective was to increase productivity, vigorousness, and adaptability to local conditions, which enable the exploitation of the advantage of dominant gene effects. Hybridization in coffee primarily considers the mating systems of Arabica (inbreeding) and Robusta (outbreeding) coffee [54]. Geneti [30] explained that different sets of crosses were made between parental lines selected for desirable traits such as yield, resistance to coffee berry disease and coffee leaf rust, quality, vigor, and others through heterosis and combining ability analysis.

#### Coffee Heterosis in Ethiopia

Research results in Ethiopia indicated that the effect of heterosis is limited and not as yet at the required amount [72]. The perennial nature of the crop is a great challenge as it takes several years to obtain significant results [73]. However, recent studies on several sets of crosses have shown the presence of a considerable amount of heterosis in crosses among indigenous cultivars and the potential to improve coffee yield through hybridization of plants originally collected from similar or different regions [23].

In Arabica coffee, a 30–60% heterosis in yield over the best parent was observed in Ethiopia [74]. Of the nine F1 hybrids only one hybrid showed negative heterosis (−8%). They also reported that the high-yielding hybrids, 7396 × F59 (Melko-CH2) and 741 × F59 (Ababuna), which have been approved for distribution to growers, showed 20% and 18% heterosis over the better parent, respectively. From 1997 to 2018, Four hybrid coffee varieties; namely, Ababuna, MCH2, Gawe, and Gera were released. These hybrids give 24–2.6 tones/ha clean coffee in the medium-altitude coffee-growing areas of southwest Ethiopia; whereas the best control and parent, Dessu, provided 1.82 tons/ha [75].

Mohammed [76] also explained that coffee hybrids produced from the distant parents, both in origin and in growth behavior, differ the most heterotic for coffee yield. Another group of hybrids that have distant parental origins but similar growth habits are the second most heterotic hybrids. For hybrids with both parents of similar origin and development, the behavior has the lowest magnitude of heterosis. As a result, increasing parental distance was strongly linked to increased heterosis.

#### Combining Ability

Combining ability evaluations for yield and morphological features in coffee revealed that both additive and non-additive gene activities have a role in their inheritance [26].

However, the non-additive gene activity was more important than the additive components. Although not broad, studies on combining ability have been undertaken in Ethiopia.

Ayano et al. [77] investigated the capacity of five parental lines from southwestern Ethiopia to combine yield and morphological features in diallel crosses. The mean squares for both GCA (general combining ability) and SCA (specific combining ability) effects on across location analysis were highly significant for yield, indicating that both additive and non-additive effects on yield were present. The significant percentage contribution of SCA over GCA, on the other hand, may indicate that non-additive gene activity predominates. Fruit length, fruit width, fruit thickness, bean length, bean width, bean thickness, and 100 bean weight were all found to be controlled by both additive and non-additive gene activities.

Mohammed [76] investigated the combining ability for coffee quality in diallel crosses among five parents from southwestern Ethiopia (Kaffa type) and southeastern Ethiopia (Sidamo type). Two hybrids, viz. 7440 × 75,227 and 744 × 1681, were evaluated. Sidamo coffee quality was the best specific combination for all coffee quality parameters. The results also indicate that non-additive gene actions are important for acidity, body, cup quality, and overall quality and additive gene actions for flavor. Dula et al. [78] also observed that some crosses that involved parents with good general combiners were poor specific combinations. This result indicates that parents with high general combining ability effects (GCA) are not always able to produce crosses with high specific combining ability effects (SCA). In addition, parents with a negative general combining ability effect might not always produce crosses with low specific combining ability effects for all the agronomic traits of interest.

Development of Pure Line Varieties

Initially, the pure-line variety development program was mainly focused on the releasing of varieties adapted to the agronomic characteristics of low to high altitude of all coffee growing regions. As a result of this effort, from 1978 to 2018 a total of 37 purebred varieties, including CBD-resistant varieties developed under the crash plant breeding program (Table 3), were released and distributed for production in different regions of Ethiopia (both low and high altitude) where coffee is mainly produced. However, incognizant of the long time required to develop improved varieties through the conventional breeding approach and the ever-increasing demand for quality and traceability, the coffee breeding program was focused on developing improved varieties for producing coffee with a unique flavor. Accordingly, the 'Local landrace variety development' breeding strategy, which is described by Bayetta and Pierre [26], was designed and has been under implementation since 1994. After critical field and laboratory evaluation for yield, diseases, and quality, 11 new local landrace varieties were released for the Hararge, Sidam/yirgacheffe, and Wollega coffee-producing areas in 2010 [79]. In addition, large numbers of promising selections from different batches of collections are also under evaluation at different trial locations. Afterward, the selected varieties will be verified and released. So, selection would have a significant role in broadening the genetic base of the improved varieties.

4.1.3. Biotechnology in Breeding

Coffee breeding by conventional methods is a long process involving several techniques, namely, the selection from wild populations followed by hybridization and progeny evaluation, backcrossing, and interspecific crosses. Unfortunately, these traditional methods of improvement are very slow (it takes over 30 years to obtain a new cultivar using any of these methods) and expensive and the resulting seed production and distribution are insufficient to satisfy the needs of coffee growers [55,80]. In Ethiopia, there are biotechnology activities started by National Agricultural Biotechnology Laboratory and regional tissue culture laboratories. Even if plant tissue culture biotechnology is at the initial stage, it has been given the highest priority in biotechnological research and development. Cof-

fee propagation by tissue culture was conducted on hybrid coffee varieties by using leaf micropropagation [81].

**Table 3.** Released coffee varieties/hybrids in Ethiopia.

| Variety Group | Number of Varieties | Year of Release | Yield Range (q/ha) | |
|---|---|---|---|---|
| | | | On Station | On Farm |
| CBD Resistant Varieties | 16 | 1978–2018 | 12.2–23.8 | 6.0–10.0 |
| Low and Midland Varieties | 5 | 1997–2002 | 16.6–23.4 | 9.0–20.0 |
| Hybrid Varieties | 4 | 1997–2018 | 24.0–26.0 | 13.0–20.0 |
| Highland Varieties | 4 | 2006 | 16.4–23.5 | 15.2–16.2 |
| Land-race Varieties | 12 | 2006, 2010 | 11.9–20.4 | 7.2–16.2 |
| **Total** | **41** | | | |

Source: EIAR [54].

According to JARC and EIAR [68] in Ethiopia three different methods can be used for the propagation of coffee plants through tissue culture: microcuttings, direct somatic embryogenesis, or plant regeneration from callus (indirect somatic embryogenesis). The crop can also be propagated in vitro through embryo culture and anther-culture. Indirect somatic embryogenesis from leaf pieces has so far been proved to be the best and most efficient means of coffee micropropagation due to the possibility of using liquid media, either using shakers or bioreactors, rather than using semi-solid materials.

Workia et al. [82] conducted their study on the somatic embryogenesis of a coffee hybrid (Aba Buna) by using leaf explants in Holeta agricultural research center. Their results indicate that the maximum number of roots per plantlet ($3.0 \pm 1.0$) was obtained on a half-strength MS medium containing 0.5 mg $L^{-1}$ indole-3-butyric acid (IBA). The acclimatization of plantlets was achieved with a survival rate of 96.9%. This finding was very important and a great achievement since the most challenging feature of coffee tissue culture in Ethiopia was the acclimatization of embryo-derived plantlets.

There are a few studies on the molecular genetic diversity of Ethiopia coffee using markers such as RAPD, RAPD, AFLP, ISSR, and microsatellite (SSR) [83,84], which were restricted to germplasm from the southwestern part of the country. However, each coffee-producing region has its own unique germplasm which should be conserved independently to be used for future breeding efforts.

On the other hand, several DNA-based techniques were used in different studies about coffee genetics. These include RAPD, AFLP, ISSR, and microsatellite (SSR) markers [68,69,83–89]. Among the several types of molecular markers, microsatellites (SSR) are most commonly used in genetic diversity studies because they offer several advantages, including the high degree of polymorphism, repeatability, reproducibility, codominance, technical simplicity, speed, and multiallelism [90]. Similarly, Tadesse et al. [91] conducted a study on 42 EIAR released varieties that were assessed by using 14 simple sequence repeat (SSR) markers (Table 4). The result showed polymorphism among the varieties. They found a high average number of polymorphic alleles (7.5) and polymorphic information content (PIC = 80%) per locus. They also explained the highly polymorphic nature of the SSRs used and revealed the presence of a high level of genetic diversity among the commercial Arabica coffee varieties grown in Ethiopia.

**Table 4.** Jaccard's genetic similarity coefficient among pairwise combinations of local landrace varieties from different geographic origins using 14 (SSR) markers.

| Origin | Variety Name | 1 | 2 | 3 | 4 | 5 | 6 | 7 | 8 | 9 | 10 | 11 | 12 |
|---|---|---|---|---|---|---|---|---|---|---|---|---|---|
| South Ethiopia | SR1(1) | 1.00 | | | | | | | | | | | |
| | SR2(2) | 0.18 | 1.00 | | | | | | | | | | |
| | SR3(3) | 0.28 | 0.42 | 1.00 | | | | | | | | | |
| | SR4(4) | 0.37 | 0.34 | 0.39 | 1.00 | | | | | | | | |
| East Ethiopia | HR1(5) | 0.29 | 0.43 | 0.54 | 0.40 | 1.00 | | | | | | | |
| | HR2(6) | 0.38 | 0.39 | 0.37 | 0.33 | 0.44 | 1.00 | | | | | | |
| | HR3(7) | 0.37 | 0.24 | 0.33 | 0.35 | 0.4 | 0.6 | 1.00 | | | | | |
| | HR4(8) | 0.49 | 0.33 | 0.40 | 0.47 | 0.51 | 0.45 | 0.51 | 1.00 | | | | |
| West Ethiopia | WR1(9) | 0.42 | 0.29 | 0.28 | 0.33 | 0.38 | 0.35 | 0.41 | 0.38 | 1.00 | | | |
| | WR2(10) | 0.48 | 0.32 | 0.30 | 0.36 | 0.31 | 0.31 | 0.33 | 0.37 | 0.57 | 1.00 | | |
| | WR3(11) | 0.34 | 0.28 | 0.33 | 0.39 | 0.40 | 0.34 | 0.29 | 0.31 | 0.42 | 0.44 | 1.00 | |
| | WR4(12) | 0.31 | 0.32 | 0.30 | 0.30 | 0.37 | 0.34 | 0.27 | 0.31 | 0.38 | 0.54 | 0.47 | 1.00 |

Source: Tadesse et al. [91].

The genetic association between 28 *Coffea arabica* genotypes from Ethiopia was also assessed by Dessalegn et al. [92] by using 10 Amplified Fragment Length Polymorphism (AFLP) primer combinations and the result indicated that the level of polymorphism was 30.9% and 18 markers were unique to 10 genotypes. Similar results were also reported by Gudeta [77,93].

The tissue culture technique for coffee is essential for mass and rapid propagation, and for the production of disease-free seedlings. The coffee protocol was optimized for tissue culture activity and a large number of tissue culture plantlets distributed to farmers in various parts of the country, which is a major achievement of tissue culture breeding in Ethiopia [25].

### 4.2. Main Breeding Objectives

In Ethiopia, coffee research started in 1967, later than in other countries, and the actual breeding program was initiated in 1978 [70,94,95]. In Ethiopia, the main objectives for Arabica coffee breeding programs are developing new and high yielders, good quality, disease- and pest-resistance (primarily to CBD), high planting density, and widely adaptable cultivars that can be used across all coffee growing regions of the country. To achieve these goals, breeding strategies have been directed towards identifying high-yielder from populations for the development of improved purebred cultivars and crosses between higher-yield plants to develop hybrids. To date, 41 high-yielders and disease-resistant coffee cultivars have been released to growers [96].

### 4.2.1. Breeding for Productivity/Yield

The identification of superior parental lines which have high yielding potential and desirable traits is the basis of the coffee breeding program. Coffee breeders apply various methods of breeding, including conventional and molecular breeding for selection. Some coffee breeding centers are now emphasizing hybrid coffee varieties as the best strategy to increase plant productivity rapidly. Coffee hybrids have also been found to have greater yield stability over location and time (fewer genotypes × environment interaction effects). The chances of substantial (transgressive) hybrid vigor are increased by combining parents selected from genetically divert subpopulations, such as a cross between common cultivars and Ethiopia accessions [68].

Recent research results show the importance of Ethiopian coffee genetic material in breeding programs for high yield and disease resistance [97]. The Ethiopian *C. arabica* lines were used as parents and crossed with commercial varieties to achieve strong hybrid vigor, resulting in over 30% higher productivity of the F1 hybrids in Central America [68].

### 4.2.2. Breeding for Diseases Resistance

Coffeeberry disease (CBD) and coffee leaf rust (CLR) are the two most serious and destructive diseases of Arabica coffee. Unlike CBD, which is confined to the African continent, primarily the highlands of East Africa, CLR is a generalized epidemic occurring in virtually all coffee-growing regions of the world. The within-pathogen variability in *Hemileia vastatrix* is huge, making resistance breeding complex. So far, more than 30 physiological races have been identified. According to Carvalho [36], altogether a minimum of seven dominant major genes have been identified for resistance and seven recessive genes have been expected to occur in the pathogen that nullifies the corresponding inheritable factor of the coffee plant for resistance. Currently, a hybrid known as Catimor (Caturra × Hybrido de Timor), with almost complete resistance to almost all races has been developed and is under production in some countries. Ethiopian coffee collections have been used to expand the genetic base of cultivated varieties and increase resistance to the nematode Meloidogyne incognita, a destructive and widespread pathogen of *C. arabica* in Guatemala and other coffee-producing countries.

### Breeding for Resistance to Coffee Leaf Rust (CLR)

Coffee leaf rust can be inferred from its economic damage to global arabica production, estimated at USD 1–2 billion per year due to crop losses (20–25%) and the need to apply cultural and chemical control measures (10% of production cost). According to Guzzo [98], more than 75% of the coffee cultivated in the world is susceptible to the majority of physiologic pathogenic races. Most breeding programs are aiming at developing a varieties resistance to this disease, particularly in Ethiopia [99].

In Ethiopia, the large genetic diversity of Arabica coffee, the high level of horizontal (non-specific) resistance to this disease, and the availability of at least some incomplete resistance might likely protect coffee against rust under prevailing conditions. Partially leaf rust-resistant coffee genotypes were identified from lowland forest coffee of southwestern Ethiopia. The presence of such a wide range of resistances to leaf rust of coffee in the wild forest population provides an opportunity to develop and use resistant materials for coffee leaf rust management, but this is as yet unexploited [100].

### Breeding for Coffee Berry Disease Resistance

Coffee-berry disease (CBD) is anthracnose of the green and ripening berries caused by the fungus *Colletotrichum kahawai* sp. nov. It was first detected in 1922 in the Sotik area, South of Mt Elgon in Western Kenya, on newly established plantations of imported Arabica coffee with a narrow genetic base. CBD is the major factor threatening coffee production in Ethiopia, Kenya, Tanzania, and other African countries. Since the disease affects the harvestable berries, it causes direct yield loss but does not influence the vegetative vigor or subsequent production potential of the plant. In Ethiopia, the national average yield loss due to the disease is estimated at 20 to 25%. However, the loss may reach 100% during favorable seasons in some areas where altitude and rainfall are high. The cost of fungicides is, however, high and it is estimated to cost ETB 150 million (USD 30 million) per year to spray all the coffee farms in Ethiopia [101]. This cost was reported to be prohibitive for the small farmers who manage 95% of the total of 320,000 hectares in Ethiopia.

The breeding programs started about 30 years ago in Ethiopia have succeeded in developing new cultivars with long-term CBD resistance. Farmers' acceptance of CBD-resistant hybrid varieties such as Ababuna is higher than of the earlier released lines, because of better agronomic performance, but data on the actual planted area of CBD-resistant cultivars are unavailable [40]. Figure 3 illustrates the generalized outlines of the breeding scheme that was proposed for the Arabica coffee breeding program in Ethiopia.

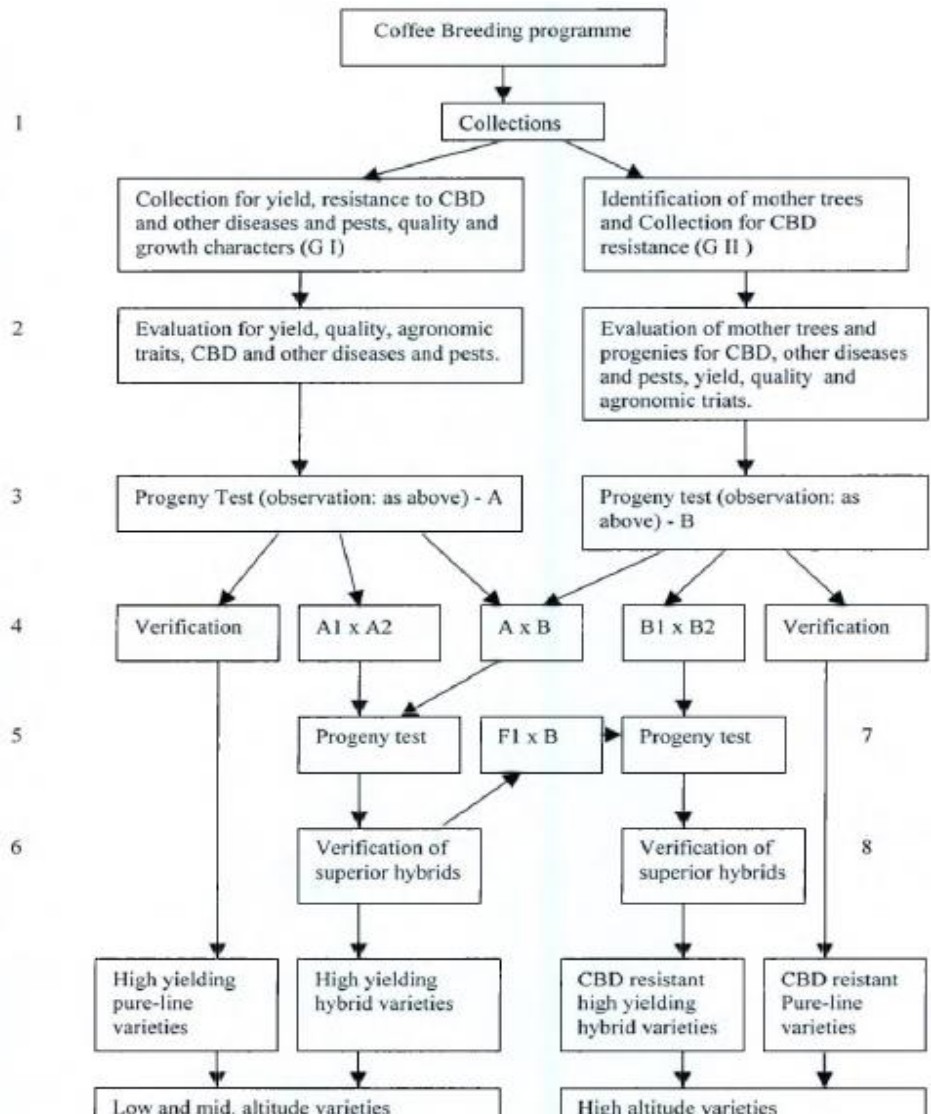

**Figure 3.** A generalized outline the breeding scheme was proposed for the Arabica coffee breeding program at JARC. Source: Admikew [102].

Breeding for Coffee Wilt Disease Resistance

Resistance to wilt disease depends in part on coffee plant genetic potential for virulence within the pathogen populations, the inoculum's concentration, and the genetic potential of the host [103]. The resistance of a plant (or tissue) changes sequentially during growth and development; thus, certain growth stages are more favorable than others for the comparison of resistant and susceptible cultivars. Attempts to control CWD are fundamentally based on the breeding of resistant plants, environmental management, and synthetic fungicide application [104].

Twelve Arabica coffee genotypes were tested from diverse agroecological zones with different resistance reactions (resistant, moderately resistant, and susceptible) based on artificial inoculation tests or natural CWD infested soils to verify previous results and select promising resistant genotypes (Figure 4). The result indicated that genotypes 279/71, 971, 974, and 79,233 exhibited low wilted seedling percentage or promising resistant genotypes, the minimum number of defoliated leaves, and extended incubation period compared to other genotypes. Therefore, these genotypes are important for inclusion in future resistance breeding programs [105].

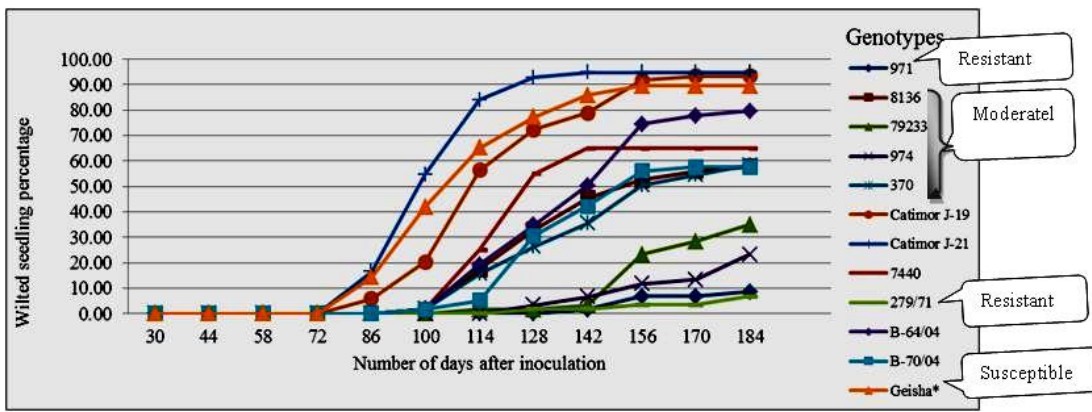

**Figure 4.** Arabica coffee genotypes wilted % progress on-time trend in artificial inoculation. Source: Admikew [102].

In Ethiopia, most of the released Arabica coffee cultivars are susceptible to coffee wilt disease [106]. However, among the released varieties Sidama or Yirgachefe varieties; *Fayate* and *Odicha* are found to be high to moderate resistant [107]. According to Demelash [108], coffee genotypes 370 and 279/71 are found as resistances as compared with the check (*catimor-19*). Additionally, cultivar Merdacheriko is considered moderately resistant in the greenhouse tests [109]. A future scheme for developing coffee wilt-resistant *Coffea arabica* varieties was indicated in Figure 5.

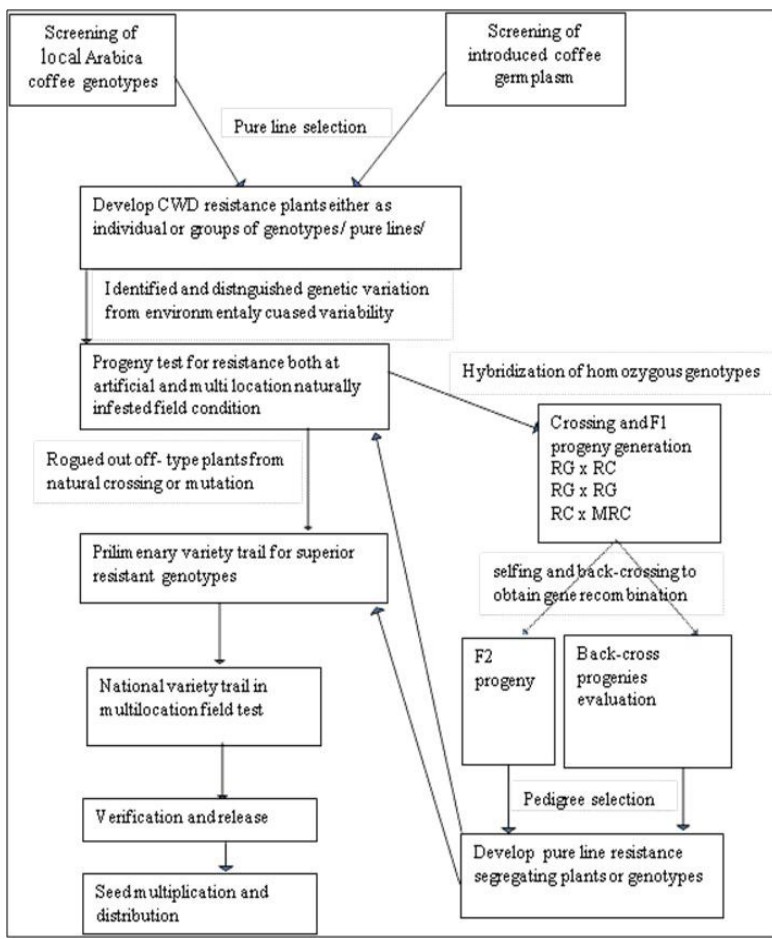

**Figure 5.** Future scheme for developing CWD resistant *Coffea arabica* varieties; RG = resistant genotype, RC = Resistant cultivar, and MRC = moderately resistant cultivar. Source: Bayetta [26].

### 4.2.3. Breeding for Quality

Selection for bean and cup quality of coffee, in general, has received a lot of attention in the selection of Arabica coffee, especially in countries that produce light (washed) coffee. This was due to the fact that the quality of new disease-resistant cultivars should be at least equal to that of the traditional cultivars to uphold the country's reputation and position in the world coffee market. Most components of coffee quality show considerable (additive) genetic variation, but they are also affected by environmental factors [55].

Among the genotype of Arabica coffee, three groups of plants can be identified: the wild genotypes from the Sudan–Ethiopian region; the cultivated non-introgressed lines (Typica and Bourbon types); and the introgressed varieties, mainly constructed from Timor hybrid genotypes. Arabica coffee is considered to be a good quality coffee (2n = 2x = 44) [110]. Abeyot et al. [111] conducted a study on the variability and association of quality and biochemical attributes in some promising *Coffea arabica* germplasm collections in southwestern Ethiopia. The results indicated that the coffee collections from Sheko, Dizi, and Meanit exhibited large differences in sensory and biochemical characteristics compared to collections from other origins. They also found that good cup quality parameters were inversely related to caffeine, bitterness, and astringency at the phenotypic level, suggesting that coffee breeding strategies within and between geographic regions may improve quality with its known provenance record.

### 5. Coffee Research Achievements in Ethiopia

Considerable research and extension efforts have been exerted on coffee by focusing on developing high yielding, CBD-resistant, and widely adaptable to different agroecologies. Several (41–37 pure lines and 4 hybrids) improved coffee varieties were released and have been disseminated for production in different coffee growing zones (Table 5) [23].

**Table 5.** Released coffee varieties registered in Ethiopia from 1986–2018.

| NO. | Variety/Cultivar | Year of Released | Yield(q/ha) | | Canopy | Breeder/Maintainer |
|-----|------------------|------------------|-------------|---------|--------|--------------------|
| | | | On Station | On Farm | | |
| 1 | 741 | 1977/78 | 12.2 | 6–7 | Open | JARC/EIAR |
| 2 | 744 | 1979/80 | 16.6 | 8–9 | Open | JARC/EIAR |
| 3 | 7440 | 1979/80 | 16.2 | 8–9 | Intermediate | JARC/EIAR |
| 4 | 7454 | 1980/81 | 18.3 | 8–9 | Intermediate | JARC/EIAR |
| 5 | 7487 | 1980/81 | 23.8 | 9–10 | Intermediate | JARC/EIAR |
| 6 | 74,110 | 1978/79 | 19.1 | 9–10 | Compact | JARC/EIAR |
| 7 | 74,112 | 1978/79 | 18.1 | 9–10 | Compact | JARC/EIAR |
| 8 | 74,140 | 1978/79 | 19.7 | 9–10 | Compact | JARC/EIAR |
| 9 | 74,148 | 1979/80 | 18.0 | 6–7 | Compact | JARC/EIAR |
| 10 | 74,158 | 1978/79 | 19.1 | 9–10 | Compact | JARC/EIAR |
| 11 | 74,165 | 1978/79 | 17.3 | 8–9 | Compact | JARC/EIAR |
| 12 | 754 | 1980/81 | 14.8 | 7–8 | Compact | JARC/EIAR |
| 13 | 75,227 | 1980/81 | 17.9 | 8–9 | Open | JARC/EIAR |
| 14 | Dessu | 1982 | 20.0 | | Open | JARC/EIAR |
| 15 | Ababuna (741 × Dessu) * | 1997 | 23.8 | 15.5 | Open | JARC/EIAR |
| 16 | Melko-CH$_2$ (7395 × Dessu) * | 1997 | 24.0 | 13.1 | Intermediate | JARC/EIAR |
| 17 | Catimor J-19 | 1997 | | | | JARC/EIAR |
| 18 | Catimor J-21 | 1997 | | | | JARC/EIAR |
| 19 | Gesiha | 2002 | | | | JARC/EIAR |
| 20 | Me'oftu | 2002 | | | | JARC/EIAR |
| 21 | Gawe * | 2002 | | | | JARC/EIAR |
| 22 | Angafa | 2006 | | | | JARC/EIAR |
| 23 | Yachi | 2006 | | | | JARC/EIAR |

**Table 5.** *Cont.*

| NO. | Variety/Cultivar | Year of Released | Yield(q/ha) | | Canopy | Breeder/Maintainer |
|-----|-----|-----|-----|-----|-----|-----|
| | | | On Station | On Farm | | |
| 24 | Buno-washi 2–05 (7416) | 2006 | | | | JARC/EIAR |
| 25 | Mocha (H-739/98) | 2010 | | | | JARC/EIAR/MCARC/OARI |
| 26 | Bultum (H-857/98) | 2010 | | | | JARC/EIAR/MCARC, ORARI |
| 27 | Mocha (H-739/98) | 2010 | | | | JARC/EIAR/MCARC/OARI |
| 28 | Mercha-1 (H-823/98) | 2010 | | | | JARC/EIAR/MCARC, OARI |
| 29 | Harusa (H-674/98) | 2010 | | | | EIAR/MCARC/OARI |
| 30 | Menesibu (W78/84) | 2010 | | | | JARC/EIAR |
| 31 | Sende (W92/98) | 2010 | | | | JARC/EIAR |
| 32 | Challa (W76/98) | 2010 | | | | JARC/EIAR |
| 33 | Haru-1(W66/98) | 2010 | | | | JARC/EIAR |
| 34 | Koti (85257) | 2010 | | | | JARC/EIAR |
| 35 | Odicha (974) | 2010 | | | | JARC/EIAR |
| 36 | Fayate (971) | 2010 | | | | JARC/EIAR |
| 37 | Tepi HC5 | 2016 | 20.3 | | | JARC/EIAR |
| 38 | Melko-Ibsitu | 2016 | 19.3 | | | JARC/EIAR |
| 39 | EIAR50/CH | 2016 | 20.9 | | | JARC/EIAR |
| 40 | L55/01 (Limu 1) | 2018 | | | | JARC/EIAR |
| 41 | 74,158 × 7530 (Gera coffee hybrid 1) * | 2018 | | | | JARC/EIAR |

*, hybrid varieties; EIAR = Ethiopian Institute of Agricultural Research; JARC = Jimma Agricultural Research Center; MCARC = Mechara Agricultural Research Center; CARC/OARI = Oromia Agricultural Research Institute; Source; MoA [23,112,113].

In general, the achievements made were the result of genetic diversity within the different agroecologies of the country [64]. However, the number of varieties released has been decreasing since 2010 (Figure 6).

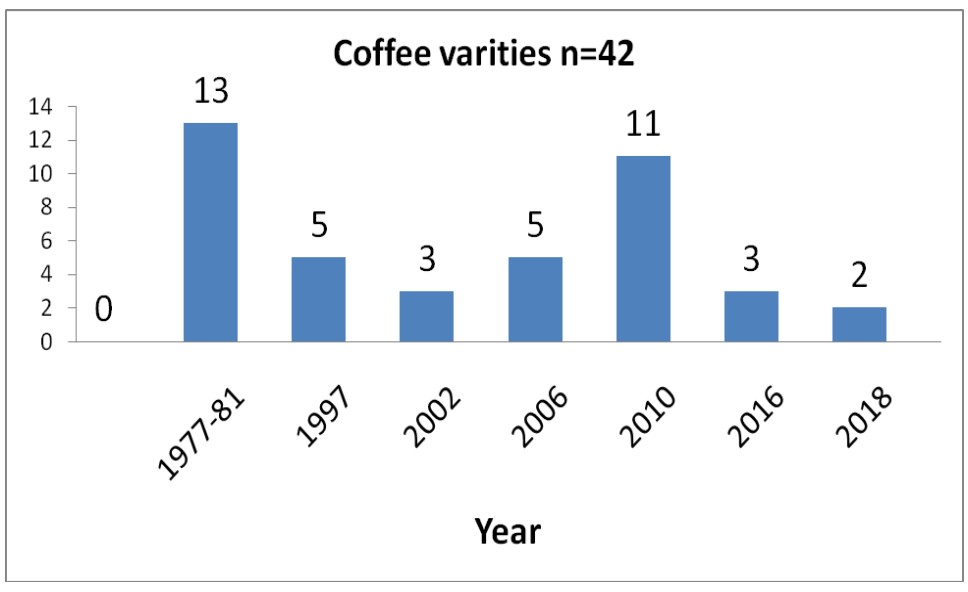

**Figure 6.** Coffee varieties were released from 1977–2018. Source; Demelash [114].

## 6. Future Strategies for Coffee Breeding

To achieve the breeding objectives, innovations have essential for improving coffee breeding. This requires the joint efforts of traditional breeding and modern genetic engineering to develop varieties adapted to climate change. Coffee genotypes are increasingly under threat due to genetic erosion [93,115]. So, attention should be paid to slowing down the process of forest degradation to reduce genetic erosion. Even though Ethiopia has broad genetic bases, this potential has not been effectively exploited due to several challenges such as limited human resources, the absence of physical facilities, and a shortage of financial resources. Participatory breeding and varietal evaluation for each specific location could be a possible approach to improve the existing conventional coffee improvement methods.

The fluctuating climate in coffee-growing areas has reduced yield and quality, increased pest and disease outbreaks, and increased production costs. Watts [116], Killeen and Harper [117], and Justin et al. [118] have declared that profoundly negative trends in the future distribution of indigenous Arabica coffee. They also argue that a 65% reduction in the number of ecologically suitable areas, and the worst-case scenario is an almost 100% reduction by 2080 due to the effect of raised global temperature. They revealed that coffee production areas might change because suitable areas become too warm or prone to periodic drought. This indicates that climate change is threatening coffee crops in virtually every major coffee-producing region of the world. Therefore, the main breeding objective should be to breed coffee varieties suitable for the changing climate.

In the field of coffee research, priorities should be given to efficient utilization and conservation of the existing genetic diversity. Varietal development through the classical breeding approach is a big challenge since it takes a long time (>15 years) to release a new variety. Therefore, it is necessary to strengthen modern coffee improvement technologies such as biotechnology (tissue culture and molecular characterization and marker-assisted selections). In this regard, researches were conducted on the application of biotechnological tools, mainly somatic embryogenesis, to propagate F1 hybrids of *Coffea arabica* and to conserve the genetic resources for future breeding programs, which show promising results [118]. Furthermore, there is a need to strengthen laws and regulations for coffee wilt disease (sanitation and quarantine) and examine whether grafting prevents the disease or not under experimentation.

## 7. Conclusions

*Coffea arabica* is an evergreen dicotyledon tree. It has sweetly smelling flowers grow in clusters in the leave axils. It is the only self-compatible species of the genus *Coffea*. However, there is 10 percent of natural cross-pollination which could be high enough to retain a certain level of heterozygosity in the population.

Breeding methods applied to depend on the type and the source of traits improved and the final objectives. The common methods applied include selection and hybridization (intra-specific hybridization), on development of high yielder and improved varieties, good cup quality, disease, and pest resistance (primarily to coffee berry disease and coffee leaf rust), high planting density, and widely adaptable cultivars on diverse agroecologies. Forty-one coffee varieties were so far developed in Ethiopia.

However, currently, the breeding method commonly applied to coffee improvement in Ethiopia is the conventional method, which is a long and slow process. In addition, climate change is threatening coffee crops in virtually every major coffee-producing region of the world. Therefore, breeding should focus to breed coffee varieties suitable for this changing climate. Thus, applications of molecular breeding techniques with conventional and indigenous knowledge could have a substantial role in improving coffee crops. It demands joint efforts of traditional breeding and modern genomic technology to develop suitable varieties for the changing climate.

**Author Contributions:** Both authors, Y.Y.M. and S.A.K., participated in preparing the review and editing the article. Both authors have read and agreed to the published version of the manuscript.

**Funding:** The authors received no external funding.

**Institutional Review Board Statement:** Not applicable.

**Informed Consent Statement:** Not applicable.

**Data Availability Statement:** Data are available upon request. Some data are available as a supplementary document.

**Conflicts of Interest:** The authors declare no conflict of interest.

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
