# Peer review of "Coffee (Coffea arabica L.): Methods, Objectives, and Future Strategies of Breeding in Ethiopia—Review"

_sustainability, doi:10.3390/su131910814_

Round 1

Reviewer 1 Report

The Authors carried out the study „Coffee (Coffee arabica L.) breeding methods, objectives and future strategies in Ethiopia: review” to present the previous and current studies on different breeding methods applied in Ethiopia as well as identify future strategies of breeding. The study is of relevance and general high interest to the readers of the journal. I found the paper to be overall good written. The authors made a broad and thorough literature review. However, some points should be added/changed to improve the manuscript.

My comments are as follows:

The manuscript should be prepared in accordance with the requirements of the editorial office, especially in the case of citation and the list of references.

There are many stylistic, grammatical, linguistic and text formatting errors in the whole manuscript. Moreover extensive editing of English language is necessary.

The quality of figures in very low and must be improved.

The placement of tables and figures does not correspond to the citations in the text.

The division of text into subsections is confusing and unreadable, too many subsections and repeats of information.

The coresponding author is not indicated.

I suggest to modify the title: „Coffee arabica L.: methods, objectives and future strategies of breeding in Etiophia ­– review”

Abstract is to long and it is a literal copy of other manuscript parts.

Lines 16-18: stylistically incomprehensible sentence.

Use italics for genus and species name and use single apostrophe for cultivars names consistently throughout the manuscript.

Keywords should not be repetitions of title words.

Lines 43-45: Authors must combine the two sentences into a logical one sentence.

Line 60, 70-73: Update the data about rankings.

Line 102: Coffee growth, floral biology, and fruit characters – information about fruit characters is missing in the subsection text, why?

What are the differences in your writning between trait and character? and between cultivar and variety? Cultivar and variety are not synonyms. Explain and do not mix nomenclature.

Lines 204-206: What are the differences between „selection” from Line 204 and selection from Line 206?

Lines 212-216: Combine the two sentences into a logical one sentence, avoid repetitions.

Line 237: Explain abbreviation GxE (genom x environment?).

Lines 260-261: Explain the essence of ‘crash program’ in more details.

Line 271: Delete “of coffee improvement”.

Line 335: Delete „coffee”.

Line 354: Delete „coffee”.

Lines 367-380: Authors are describing micropropagation methods used for mass production of cuttings. Examples of the use of these methods in breeding programs, and not in production, should be indicated.

Lines 411-413: Incomprehensible sentence.

Table 3: Correct „cmposit” to „composit”.

Line 416: Delete „ for coffee”.

Table 4: Add „in Ethiopia” in table title.

Lines 592-605: These are not conclusions, but repeats of text from previous chapters. Sentences should be drafted and written properly.

Author Response

Thank you for your helpful comments and suggestions. Based on comments provided we have addressed/ incorporated comments. the details of comments are uploded here.

Reviewer 2 Report

The manuscript entitled; Coffee (Coffea arabica L.) Breeding Methods, Objectives and Future Strategies in Ethiopia: Review by Yeshiwas and Asredie providing detailed information about different methodologies of coffee crop improvement in Ethiopia. However, authors did not describe why the production of Ethiopia is lower than the other high yielding countries. What approaches the other countries used to improve coffee better. Authors should include the explanation about certain advanced tools including tissue culture and marker assisted breeding and may be grafting. I will suggest including one paragraph for each of the above-mentioned aspects and reduce most of the paragraphs of conventional breeding with concise writing. Also, please check throughout the manuscript and improve the English. Furthermore, few minor comments are as below

Abstract

Please mention the key points in the abstract why we need this review and authors can mention concrete take home message based upon the current findings

Please check the reference style of this journal as it is not followed in the current version

Line 88, please replace “research and breeding effort is very limited” with “research and breeding effort are limited”

Line 97-98, please replace “breeders and/ or agronomists” with “coffee scientists”

Line 99, please replace “the objective of this review work is” with “the objectives of this review are”

Line 100-101, please rephrase both objectives to express the meanings

Please check throughout the manuscript and replace the “flower buds” with “floral buds”.

The author is describing selfing and crossing techniques here. Is it novel information?

Figures quality for all figures should be improved

Line 183, please replace “small holder farmers” with “small farm holders”

Table 1, please represent yield in “t/ha”

Author Response

Thank you for your helpful comments and suggestions. Based on comments provided we have addressed/ incorporated comments. detail responses are uploded here.

Round 2

Reviewer 2 Report

In my last revision I suggested following comments to be improved

"Authors did not describe why the production of Ethiopia is lower than the other high yielding countries. What approaches the other countries used to improve coffee better. Authors should include the explanation about certain advanced tools including tissue culture and marker assisted breeding and may be grafting. I will suggest including one paragraph for each of the above-mentioned aspects and reduce most of the paragraphs of conventional breeding with concise writing". 

I am happy to see that most of the minor comments are tackled well however, above-mentioned paragraph of major comments yet to be tackled. Authors completely ignored the suggestions and I can't accept the current form this manuscript 

Author Response

Thank you very much for all the given constructive comments, suggestions, and directions. We want to apologize for the forgotten and unaddressed comments.  We have updated the scripts to the required level. We have incorporated the suggested ideas, we found them very helpful and informative. 

Round 3

Reviewer 2 Report

I have no more comment but it will be ideal if authors can include a paragraph on coffee improvement through tissue culture